# Medication Use among Immigrants from Syria Living in Western Norway: A Cross-Sectional Study

George Deeb [1], Esperanza Diaz [2,3], Svein Haavik [4] and Angela Lupattelli [1,*]

[1] PharmacoEpidemiology and Drug Safety Research Group, Department of Pharmacy, Faculty of Mathematics and Natural Sciences, University of Oslo, 0316 Oslo, Norway
[2] Department of Global Public Health and Primary Care, University of Bergen, 5009 Bergen, Norway
[3] Unit for Migration and Health, Norwegian Institute of Public Health, 0473 Oslo, Norway
[4] Center for Pharmacy, Department of Clinical Science, University of Bergen, 5007 Bergen, Norway
* Correspondence: angela.lupattelli@farmasi.uio.no; Tel.: +47-22-84-55-49

**Abstract:** This cross-sectional study sought to quantify medication use and change in use of prescription-only medications purchased in the past in Syria without medical prescription versus today in Norway in an adult population originating from Syria and living in western Norway. Data on adults born in Syria and living in Norway during December 2019–January 2020 were collected via a self-administrated questionnaire in Arabic. Participants were recruited at a community pharmacy and at a refugee center. We included 148 participants (mean age 36.4 years; 38.5% females and 60.8% males) of whom 62.6% had lived in Norway for 4–6 years. Most participants had low (45.9%) or medium (39.2%) health literacy. Painkillers and analgesics were the most widely used medications, in both Norway (69.6%) and Syria (78.4%). Use of antibiotics declined significantly in Norway (31.1%) relative to Syria (65.5%); 70.9% participants used prescription-only medications in both countries, while 6.1% and 13.5%, respectively, did so only in Norway or only in Syria. This study reports a relatively high rate of medication use, particularly painkillers and analgesics both in Syria and in Norway. Participants with low health literacy reported greater use of antibiotics than those with high level in Syria but not in Norway. Use of antibiotics decreased substantially in Norway relative to the past in Syria, reaching a comparable prevalence with that in the host community. Although uncommon, prescription-only medication use only in Norway was reported by some participants.

**Keywords:** medication use; immigrant; Norway; Syria; prescription-only-medication





## 1. Introduction

Syrians have emerged as a new immigration group in Norway due to the escalating conflicts following the Arab Spring [1]. In 2010, there were very few Syrians living in Norway; by 2019 and 2022, they counted approximately 34,000 and 40,000, constituting 15% of immigrants with refugee background in the country [2,3]. Non-communicable risk factors are leading causes of low disability-adjusted life-years in Syrians, and the burden of mental health problems and medication use has increased in this population following the Arab uprisings [4].

A study conducted in 2017–2018 among early-arrived Syrian refugees in Norway showed that almost 10% of this population, especially women, used analgesics daily followed by tranquillizers, antidepressants, and sedatives [5]. However, the proportion of medication treatment among those reporting a long-term somatic disorder (e.g., cardiovascular disorder) was low, suggesting possible inadequate treatment in this population upon settlement in the host country [5]. This prior study [5] did not examine use of antibiotics specifically, and therefore knowledge about this specific medication use among immigrant from Syria in the transition to Norway remains unknown. Ensuring the continuity of health care and adequate access to treatment is crucial for integration, individual well-being, and public health at large.

The main stated goal of the health care system in Norway is equity in health care for all regardless of ethnicity, religion, language, or cultural background. Asylum seekers and refugees have the same rights to health care for physical ailments and illness, mental issues, addiction problems, and dental care as native Norwegians [6]. However, immigrants with different backgrounds have different perceptions of health-seeking behavior, signs of illness, medication-taking behavior including over-the-counter-drugs (OTC), beliefs about medicines, and health literacy, which are intuitively inherited from homeland countries. As acknowledged by the Norwegian government [7], there are several challenges within health care services related to immigrants in Norway, e.g., cultural differences, access to health information, health-seeking behavior, competence of health workforces, and research and development. These challenges are possibly even greater for people who come to Norway in adulthood than for younger generations.

A structured health care system did not exist in Syria before the war. More specifically, regarding access to and the use of medication, individuals predominantly self-diagnose and self-medicate; generally, individuals can purchase medicines in any pharmacy without a prescription, even those medications licensed as prescription-only in Western countries (e.g., antibiotics, antidiabetics, antihypertensive drugs) [8,9]. Hence, self-medication and inappropriate use of medications constitute an important health concern among the Middle East communities [10]. This situation is aggravated by the public health situation in Syria before the war, which has seen an important increase in the incidence of noncommunicable diseases such as obesity, diabetes, and coronary heart disease [4].

The current medication-taking behavior in Syrians may still be shaped by their experiences back in Syria and on their way to Norway. Understanding the extent to which Syrians use medication currently in Norway as opposed to in Syria in the past is therefore important for preventing inappropriate drug use and in parallel ensuring adequate treatment [11,12]. It is also important to map medication use in this population by sex, as women and men may have different health profiles and medication-taking behaviors. Likewise, inadequate health literacy is a common problem among some immigrant groups, and it is an important determinant of medication use and health behavior [13]. Several studies have found associations between poor literacy and poorer likelihood of self-managing health conditions, as well as poorer health outcomes [14]. Low health literacy has been found to be associated with over- or underutilization of health care services [15], which underlines the importance of examining this factor in relation to medication use.

In an adult population originating from Syria and living in western Norway, we sought to quantify and compare use and change in the use of prescription-only medications purchased in the past in Syria without medical prescription versus today in Norway with prescriptions required. Our secondary aim was to examine the use of any medication as well as OTC medications, overall and according to sex and health literacy level in the home and host country.

## 2. Results

Overall, 156 individuals completed the questionnaire, mainly in electronic format (*n* = 150, 96.2%). The response rate was 82.5% among those who were asked to participate. After quality checking the data, we excluded eight invalid or duplicate responses. We reached a final study population of 148 participants, of whom 90 (60.8%) were males and 57 (38.5%) females. The mean age of the study population was of 36.4 years (SD = 10.6). One participant did not disclose information about sex. Table 1 shows the demographic and lifestyle characteristics of the study population, overall and by sex.

Most participants were married (68.9%), had children (58.1%), and were of Arab ethnicity (82.4%). Overall, 62.6% had been living in Norway for 4–6 years, and the remainder were relatively new arrivals (3 years or less). Poor exercise habits were common in the study population, and so was being overweight or obese. Most participants were non-consumers of alcohol, especially women. The majority of the study participants (87.8%) self-rated their health as good/very good in Syria, but only 48.0% rated their health as equally good in

Norway. Only 22 participants (14.9%) had high health literacy score, while 68 (45.9%) and 58 (39.2%) had, respectively, low or medium health literacy.

**Table 1.** Sociodemographic and lifestyle characteristics of the study population, overall and by sex.

|  |  | Females (*n* = 57) | | Males (*n* = 90) | | Total (*n* = 148) [a] | |
|---|---|---|---|---|---|---|---|
|  |  | *n* | (%) | *n* | (%) | *n* | (%) |
| **Sociodemographics, lifestyle characteristics, and health literacy** | | | | | | | |
| Age (in years) | 18–25 | 12 | (21.4) | 12 | (13.3) | 24 | (16.3) |
|  | 26–40 | 27 | (48.2) | 46 | (51.1) | 74 | (50.3) |
|  | 41–52 | 14 | (25.0) | 27 | (30.0) | 41 | (27.9) |
|  | ≥53 | <5 | - | 5 | (5.6) | 8 | (5.4) |
|  | Missing | <5 | - | 0 | 0 | <5 | - |
| Mother tongue | Arabic | 52 | (91.2) | 74 | (82.2) | 127 | (85.8) |
|  | Kurdish | 5 | (8.8) | 16 | (17.8) | 21 | (14.2) |
| Ethnicity | Arab | 51 | (89.5) | 70 | (77.8) | 122 | (82.4) |
|  | Kurd | 6 | (10.5) | 18 | (20.0) | 24 | (16.2) |
|  | Missing | 0 | 0 | <5 | - | <5 | - |
| Years since arrival in Norway | ≤3 | 26 | (45.6) | 22 | (24.7) | 48 | (32.7) |
|  | 4–6 | 30 | (52.6) | 61 | (68.5) | 92 | (62.6) |
|  | ≥7 | <5 | - | 6 | (6.7) | 7 | (4.8) |
|  | Missing | 0 | 0 | <5 | - | <5 | - |
| Years of education | 1–9 (Basic) | 25 | (45.5) | 37 | (41.1) | 63 | (43.2) |
|  | 10–15 (Vocational) | 25 | (45.5) | 29 | (32.2) | 54 | (37.0) |
|  | ≥16 (College\University) | 5 | (9.1) | 24 | (26.7) | 29 | (19.9) |
|  | Missing | <5 | - | 0 | 0 | <5 | - |
| Marital status | Single | 5 | (8.8) | 34 | (37.8) | 39 | (26.4) |
|  | Married | 47 | (82.4) | 55 | (61.1) | 102 | (68.9) |
|  | Other than above | 5 | (8.8) | <5 | - | 7 | (4.7) |
| Having children | Yes | 40 | (70.2) | 45 | (50.0) | 86 | (58.1) |
|  | No | 17 | (29.8) | 45 | (50.0) | 62 | (41.9) |
| Occupational status in homeland | Working | 14 | (24.6) | 62 | (68.9) | 77 | (52.0) |
|  | Not working | 31 | (54.4) | 7 | (7.8) | 38 | (25.7) |
|  | Student | 12 | (21.0) | 21 | (23.3) | 33 | (22.3) |
| Occupational status in Norway | Working | 11 | (19.3) | 33 | (36.7) | 45 | (30.4) |
|  | Not working | 10 | (17.5) | 30 | (33.3) | 40 | (27.0) |
|  | Student | 36 | (63.2) | 27 | (30.0) | 63 | (42.6) |
| Body mass index (BMI) | Underweight | <5 | - | <5 | - | <5 | - |
|  | Normal | 16 | (29.6) | 33 | (37.1) | 49 | (34.0) |
|  | Overweight | 20 | (37.0) | 32 | (36.0) | 52 | (36.1) |
|  | Obese | 15 | (27.8) | 23 | (28.8) | 39 | (27.0) |
|  | Missing | <5 | - | <5 | - | <5 | - |
| Frequency of physical exercise | Never | 13 | (25.0) | 20 | (23.8) | 33 | (24.1) |
|  | Once or less per week | 29 | (55.8) | 37 | (44.0) | 67 | (48.9) |
|  | 2–3 times a week | 8 | (15.4) | 22 | (26.2) | 30 | (21.9) |
|  | Daily | <5 | - | 5 | (6.0) | 7 | (5.1) |
|  | Missing | 5 | (8.8) | 6 | (6.7) | 11 | (7.4) |
| Current smoking status | No, never | 30 | (56.6) | 19 | (22.1) | 49 | (35.0) |
|  | No, I quit | 7 | (13.2) | 13 | (15.1) | 21 | (15.0) |
|  | Yes, occasionally | 5 | (9.4) | 11 | (12.8) | 16 | (11.4) |
|  | Yes, daily | 11 | (20.8) | 43 | (50.0) | 54 | (38.6) |
|  | Missing | <5 | - | <5 | - | <5 | - |

**Table 1.** *Cont.*

| | | Females (*n* = 57) | | Males (*n* = 90) | | Total (*n* = 148) [a] | |
|---|---|---|---|---|---|---|---|
| | | *n* | (%) | *n* | (%) | *n* | (%) |
| **Sociodemographics, lifestyle characteristics, and health literacy** | | | | | | | |
| Current alcohol drinking habit | Never\not last year | 47 | (82.5) | 53 | (59.6) | 100 | (68.0) |
| | Weekly | <5 | - | 15 | (16.9) | 17 | (11.6) |
| | Monthly\Few times a year | 8 | (14.0) | 21 | (23.6) | 30 | (20.4) |
| | Missing | 0 | 0 | <5 | - | <5 | - |
| Health literacy | Low | 30 | (52.6) | 38 | (42.2) | 68 | (45.9) |
| | Medium | 21 | (36.9) | 36 | (40.0) | 58 | (39.2) |
| | High | 6 | (10.5) | 16 | (17.8) | 22 | (14.9) |
| **Self-rated health status** | | | | | | | |
| In Syria | Poor/very poor | 5 | (8.8) | 5 | (5.6) | 10 | (6.8) |
| | Neither | <5 | - | 5 | (5.6) | 6 | (4.1) |
| | Good/very good | 51 | (89.5) | 79 | (87.8) | 131 | (89.1) |
| | Missing | 0 | 0 | <5 | - | <5 | - |
| In Norway | Poor/very poor | 25 | (43.9) | 25 | (27.8) | 50 | (33.8) |
| | Neither | 7 | (12.3) | 11 | (12.2) | 18 | (12.2) |
| | Good/very good | 22 | (38.6) | 49 | (54.4) | 72 | (48.6) |
| | Missing | <5 | - | 5 | (5.6) | 8 | (5.4) |

Data with <5 respondents are not shown due to Norwegian data protection regulations. If "missing" is not presented, it means that the variable did not have missing values. [a] One participant did not disclose information on sex.

Most participants (84.5%) had at least once used a medication without medical prescription in Syria, while over two-thirds (77.0%) had at least once used a prescribed medicine in Norway during the last year (Table 2). Table 2 shows the differences in the proportions of prescription-only medication use among the same individuals in the past in Syria compared with the more recent time in Norway. The proportions of the most commonly used medications, such as painkillers (78.4% vs. 69.6%, −8.8% difference), antibiotics (65.5% vs. 31.1%, −34.4% difference), and gastrointestinal/digestives drugs (37.2% vs. 21.6%, −15.6% difference), declined significantly from Syria to Norway. The extent of medication use for sleeping problems increased significantly from Syria to Norway (5.4% versus 13.5%, + 8.1% difference), mainly in males. Medications for diabetes, high blood pressure, asthma, and psychotropics were used without prescription in Syria, but the proportions were low (5.4% or less, see Figure S1).

As shown in Table 3, most participants had used prescription-only medication in both Syria and Norway (70.9%). Antibiotic use only in Syria was reported by 40.5% of the participants, while 25.0% used antibiotics in both countries. Few participants (6.1%) reported prescription-only medication use only in Norway.

Figures S1 and S2 show the proportion of use of various medication groups in Syria and in Norway according to sex. Females were more likely to report use of allergy medication in the past in Syria than males (29.8% vs. 16.7%, respectively, *p* = 0.048); similar trend was observed for frequent use of prescription allergy medication more recently in Norway (15.8% among females vs. 5.6% among males, *p* = 0.048). There were no other statistically significant differences in the proportions of prescription-only medications by sex.

As shown in Table S1, the use of OTC medications in Norway was reported by 81.8% of males and 87.7% of females, independent of frequency. OTC painkillers were used frequently by almost 60% of the population, and 13–17% used them on a daily basis. There were no statistically significant differences in the proportions of OTC medications by sex.

**Table 2.** Past and current use of the most common prescription-only-medications, overall and by sex [a,b].

| Medications | Total (*n* = 148) | | | | | Females (*n* = 57) | | | | | Males (*n* = 90) | | | | |
|---|---|---|---|---|---|---|---|---|---|---|---|---|---|---|---|
| | Past in Syria | | Current in Norway | | | Past in Syria | | Current in Norway | | | Past in Syria | | Current in Norway | | |
| | *n* | % | *n* | % | % Difference (95% CI) [c] | *n* | % | *n* | % | % Difference (95% CI) [c] | *n* | % | *n* | % | % Difference (95% CI) [c] |
| Painkillers/ Analgesics | 116 | 78.4 | 103 | 69.6 | −8.8 [§] (−16.7, −0.9) | 47 | 82.5 | 42 | 73.7 | −8.8 (−19.3, 1.8) | 68 | 75.6 | 61 | 67.8 | −7.8 (−19.2, 3.7) |
| Antibiotics | 97 | 65.5 | 46 | 31.1 | −34.4 * (−44.6, −24.3) | 36 | 63.2 | 17 | 29.8 | −33.3 * (−49.9, −16.7) | 61 | 67.8 | 28 | 31.1 | −36.7 * (−49.9, −23.5) |
| Gastrointestinal/Digestive drugs | 55 | 37.2 | 32 | 21.6 | −15.6 * (−24.9, −6.1) | 20 | 35.1 | 15 | 26.3 | −8.8 (−24.5, 7.0) | 35 | 38.9 | 17 | 18.9 | −20.0 [§] (−32.3, −7.7) |
| Allergy medications | 32 | 21.6 | 22 | 14.9 | −6.7 (−15.1, 1.6) | 17 | 29.8 | 11 | 19.3 | −10.5 (−25.8, 4.7) | 15 | 16.7 | 11 | 12.2 | −4.5 (−14.7, 5.9) |
| Tranquillizers/ Sedatives | 8 | 5.4 | 11 | 7.4 | +2.0 (−3.8, 7.8) | <5 | - | 6 | 10.5 | +8.8 (−1.8, 19.3) | 7 | 7.8 | 5 | 5.6 | −2.2 (−9.5, 5.0) |
| Sleeping drugs | 8 | 5.4 | 20 | 13.5 | +8.1 [§] (1.4, 14.9) | <5 | - | 6 | 10.5 | +3.5 (−9.1, 16.1) | <5 | - | 13 | 14.4 | +10.0 [§] (2.0, 18.0) |
| Any prescription-only medication | 125 | 84.5 | 114 | 77.0 | −7.5 (−15.1, 0.3) | 48 | 84.2 | 43 | 75.4 | −8.8 (−20.6, 3.0) | 76 | 84.4 | 70 | 77.8 | −6.6 (−17.4, 4.1) |

Abbreviations: CI = Confidence interval. [a] One participant did not disclose information on sex. [b] Reported medication use with <5 respondents across groups are not shown. Only the most common medication groups are presented here (≥6 respondents in the total sample). [c] Difference in proportion of prescription-only medication use between same individuals from past in Syria to Norway. * $p$-value = < 0.001; [§] $0.05 > p$-value ≥ 0.001; $p$-values derive from the McNemar test comparing the use of prescription-only medications in Syria versus in Norway. Data with <5 respondents are not shown due to Norwegian data protection regulations.

**Table 3.** Changes in the use of the most common prescription-only medications in Syria versus Norway.

| Medications | No Prescription-Only Medication Use Neither in Syria nor in Norway | | Prescription-Only Medication Use Both in Syria and in Norway | | Prescription-Only Medication Use Only in Syria | | Prescription-Only Medication Use Only in Norway | |
|---|---|---|---|---|---|---|---|---|
| | N | % (95% CI) | n | % (95% CI) | n | % (95% CI) | n | % (95% CI) |
| Painkillers/Analgesics | 23 | 15.5 (10.5–22.4) | 94 | 63.5 (55.4–70.9) | 22 | 14.9 (10.0–21.6) | 9 | 6.1 (3.2–11.3) |
| Antibiotics | 42 | 28.4 (21.7–36.2) | 37 | 25.0 (18.6–32.7) | 60 | 40.5 (32.9–48.7) | 9 | 6.1 (3.2–11.3) |
| Gastrointestinal/ Digestive drugs | 81 | 54.7 (46.6–62.6) | 20 | 13.5 (8.9–20.1) | 35 | 23.6 (17.4–31.2) | <5 | - |
| Allergy medications | 104 | 70.3 (62.4–77.1) | 10 | 6.8 (3.7–12.2) | 22 | 14.9 (10.0–21.6) | <5 | - |
| Any prescription-only medication | 14 | 9.5 (5.7–15.4) | 105 | 70.9 (63.1–77.7) | 20 | 13.5 (8.9–20.1) | 9 | 6.1 (3.2–11.3) |

Abbreviations: CI = Confidence interval. Only the most common medication groups are presented here (≥6 respondents in the total sample). Data with <5 respondents are not shown due to Norwegian data protection regulations.

Figures S3 and S4 show the proportions of the most commonly used prescription-only medication groups respectively in Syria and in Norway, respectively, according to level of health literacy. There was a statistically significant difference between use of any prescription-only medication without medical prescription in Syria in individuals with low health literacy compared with those with high health literacy (95.6% versus 77.3%, *p*-value: 0.002). This difference was also observed in relation to the use of antibiotics without medical prescription in Syria: 80.9% in individuals with low health literacy versus 50.0% in those with a high level (*p*-value: 0.001). In Norway, the use of any prescription medication did not significantly differ according to level of health literacy, and nor did the use of antibiotics (22.7% among individuals with high health literacy versus 30.9% in those with low, *p*-value: 0.597). As shown in Figure S5, the use of OTC drugs in Norway did not significantly differ across groups with varying level of health literacy.

### 3. Discussion

This study adds to the limited available literature on medication use among Syrian immigrants currently living in Norway. Overall, we found that 8 out of 10 participants reported use of at least one prescription-only medication but without medical prescription, previously in their homeland. In Norway, the recent use of any prescribed medication was lower than in Syria irrespective of sex, but use in both countries remained common. However, in both Syria and Norway, females reported a significantly greater use of allergy medications than males, and this was the sole medication group for which sex differences were observed. The reported use of prescription-only medication without medical prescription in Syria, specifically some antibiotics, was higher among individuals with low health literacy compared with individuals with high health literacy, but this result did not emerge in Norway.

Painkillers and analgesics comprised the most widely used medications in both Syria and Norway, and their use in both countries was common. Furthermore, 6.1% of the participants reported this medication use only in Norway, and these were mainly males. These findings may reflect a higher burden of long-term pain experiences among the study participants. A recent study examined the health status and the use of medication among Syrian refugees in two different migration phases: in a transition setting (Lebanon) and in a recipient country (Norway) [5]. This study showed that almost one-third of Syrians refugees were struggling with chronic pain, with 46% and 57% reporting headaches and musculoskeletal pain, respectively [5]. A strong relationship between trauma exposure and chronic pain among Syrian refugees was found in the Norwegian CHART study [16]. Although our study did not specifically ask about the type of analgesic or painkillers used and the dosages, use of specific medications (e.g., NSAIDs) should be not protracted for long periods at high dosages [17,18].

No sex-specific differences were observed in relation to prescribed medications for treatment of chronic diseases, except that females reported greater use of allergy medications than males in both Syria and Norway. This finding may reflect a greater disease burden in women, which has been identified in prior research [19]. Our estimated proportions of prescribed medications for the treatment of diabetes, high blood pressure, and high cholesterol were relatively low but higher in Norway than in Syria. Results of the CHART study have shown that a large number of Syrian refugees with noncommunicable disorders do not receive adequate treatment with medications [5]. However, as we did not measure the burden of various disorders in our study, we cannot corroborate or refute whether Syrians may be undertreated in the host country. At the same time, the higher proportion of prescription-only medication users for chronic diseases in Norway, relative to Syria may suggest a greater disease burden of Syrians once they settle in a new country. However, alternative explanations are possible: a higher age and different treatment practices in the hosting country relative to Syria or how medication use was measured in the two settings. Indeed, while participants were asked about ever use of prescription-only medication bought without a regular prescription in Syria, for Norway, the question pertained to

prescription-only medication use bought with a prescription during the last 12 months. Thus, we may have underestimated past use of prescription-only medication after medical prescription in Syria. Our estimates of drug use were also higher than in CHART in relation to the treatment of long-term disorders such as high blood pressure (8.7% in our study vs. about 4% in CHART), high cholesterol (5.5% vs. 1–2%), or diabetes (type 1: 5.2%, type 2: 6.6% vs. 2–3%) [5]. The recruitment of participants at a community pharmacy may have inflated the number of recent medication users in our study. Furthermore, the low sample size of our study may have produced more unstable estimates of medication use.

The use of antibiotics and gastrointestinal/digestive medications declined significantly from the past in Syria to more recently in Norway, and a substantial proportion of participants had used these medication groups only in their homeland. This is an important finding as the recent use of prescribed antibiotics in Syrians living in Norway reached a comparable rate as that for the host community [20]. This may suggest that once established in the hosting country, Syrian immigrants generally use prescribed antibiotics and gastrointestinal/digestive medications to a comparable extent as Norwegians. In our study, the use of antibiotics without medical prescription in Syria was significantly greater among participants with low health literacy compared with those with high levels; however, this difference by health literacy level did not manifest for prescription-only or OTC medication use in the hosting country. As shown in prior research [21], individuals with low health literacy are more likely to have limited understanding of the proper use and associated side effects of medications. This factor, combined with the lack of a structured health care system in Syria versus the presence of a structured and regulated system in Norway, may explain these results. Recent national data for Norway [22] have shown that immigrants are surprisingly well informed about the structure of the Norwegian health service, but those with lower educational or socioeconomic background face important health literacy and communication challenges with health care professionals. Taken together, this underlines the importance of tailored patient education strategies to improve understanding of health information among immigrants, which can in turn improve outcomes and avoid unnecessary medication use [22].

In line with prior findings for this population in the CHART study [5], psychotropic drugs were more commonly used in Norway than in the homeland, and in specific, we observed a significant increase in the use of medications for sleeping problems, mainly among males. The greater use of these medications after arrival in Norway may be a marker of distress in this population; indeed, a substantial proportion of our study population rated their health as poor/very poor in Norway. The observed decline in self-rated health in Norway relative to the past in Syria might be explained by psychological factors e.g., distress [23], sociodemographic factors like increasing age and change in occupational status [24,25], or even changes in BMI and physical activity [26], and lastly but most importantly, social capital [27]. However, due to the small sample size, we could not examine what individual factors were related to the initiation of drug treatment for sleeping problems in Norway. Our findings on the use of antidepressants and hypnotics in males versus females are partly in line with results from a population-based study in Sweden and filled prescription data in Norway [28,29].

*Strengths and Limitations*

One strength is the relatively high response rate (82.5%) among participants who were asked to participate. However, we could not calculate a standard response rate due to the impossibility of counting how many individuals read the study brochure. Providing written information about the study translated into Arabic and interacting verbally in the participant's own language was the main facilitation for this rate of response. Most of the questions in the questionnaire were simple and easy to understand by the study population, and they had been used in prior research in Norway. The study collected data at different opening hours and days at the pharmacy, which limited the risk of selection of specific

patient groups. We used a web-based approach coupled with paper-and-pencil to increase participation in the study.

Several limitations are worth mentioning. Firstly, as the study was based on a self-administered questionnaire, participants may have over- or underreported certain behaviors. Secondly, participants were asked to report several events that had occurred years ago in their homeland, which may increase the risk of poor recall. However, we were conscious to simplify all questions directed to participants by including written explanation about the concepts of each question and providing well-known examples to enhance recall. In addition, in the questions related to past medication use, participants had the option to answer "I do not remember", so we assume low influence on the overall results. Our question about medication use and its frequency in Norway was not completely equal to the one concerning the past in Syria. For Syria, participants were asked about prescription-only medications bought without prescription ever in life, while in Norway, the question pertained only to prescription-only medication bought with a prescription during the last 12 months. Our observed differences in the proportions of prescription-only medication use in the Syria–Norway comparison may partly be attributable to these factors rather than to the refugee situation. One important limitation of this study is the small sample size; the low number of participants impeded us in more closely exploring the use of those medications that are less commonly used (e.g., drugs for diabetes, psychotropics), as well as in examining predictors of change in prescription-only medication from Syria to Norway. Our population was partly recruited at a pharmacy and included a substantial proportion of individuals who were overweight and obese and who had low physical activity; being a pharmacy customer could increase the risk of having poorer health and higher needs for medication use. Therefore, our results may not be fully generalizable to the healthier segment of the Syrian population living in western Norway. Furthermore, the interpretation and generalizability of our results must be carefully considered in view of the small number of participants recruited.

## 4. Materials and Methods

### 4.1. Study Design and Data Collection

This is a cross-sectional study conducted between December 2019 and January 2020 in western Norway, Øygarden municipality, at two different sites: (i) the service center for refugees and (ii) a centrally located community pharmacy. Individuals age over 18 years, who were born in Syria and now residing in Norway could participate. There was no exclusion criterion. Eligible participants were invited to participate. At the community pharmacy, one member of the research team able to speak Arabic recruited individuals at different opening hours for at least eight hours a day five days a week. Participants had the option to fill out an electronic or paper-and-pencil questionnaire. We applied two methods for recruitment: (i) direct inquiry to eligible individuals to participate in the study, with distribution of paper and pencil or link (or QR code) to the electronic questionnaire; (ii) self-selection into the study by reading the study brochure at both sites and accessing the electronic questionnaire via scanning the QR code on the study brochure.

Based on the wide range of prevalence estimates for more common prescription-only medication and OTC drug use in prior research in Norway [29] and Syria [30], we assumed that between 15% and 80% of individuals had at least once used a medication in the last 12 months or in the past. The required sample size to provide proportions with 6–8% precision under different scenarios within the above range of prevalence estimate was 137–151 individuals.

### 4.2. Measurements

A structured, self-administered questionnaire was used to collect data on medication use and additional characteristics, behaviors, and conditions. The questionnaire was first developed in English and then translated into Arabic by a native Arabic speaker from Syria, and it took approximately 15–20 min to complete (the full questionnaire is available as

Supplement S1). The electronic format of the questionnaire was available via Nettskjema within the University of Oslo.

Participants were first asked to self-report which medications they had been taking in their home country without medical prescription, within a predefined list of 12 medication groups (e.g., medicines for high cholesterol, medicine for diabetes mellitus type I, painkillers/analgesics). To enhance recall, examples of brand name products were provided for each medication group. The 12 medication groups included drugs that are generally prescription-only in Western countries. Participants were then asked to report their recent use of prescription-only medications in Norway, along with their frequency, within the same predefined list of 12 medication groups, by answering the question: "Have you in the last 12 months used any of the following prescription-only-medicines (prescription-only medication)?" For prescription-only medication use in Norway, we aggregated the different use frequency categories into a single group. In addition, the study measured use of OTC medications in Norway within a pre-defined list of seven medication groups. The above questions have been adapted from the HUNT3 study [31] and from prior research conducted in Norway [32]. Based on prescription-only medication use status in Syria and Norway, we defined four mutually exclusive groups: (i) no prescription-only medication use in either Syria or in Norway; (ii) prescription-only medication use both in Syria and in Norway; (iii) prescription-only medication use only in Syria; and (iv) prescription-only medication use only in Norway. This categorization was performed for any prescription-only medication, as well as for most commonly used medication groups.

Past self-rated health in Syria was measured by the following question: "When you were living in your home country, how did you consider your health? (You may think about the time just before departing for Norway)". Current health in Norway was measured by a similar question. These were adapted from a validated measurement tool that showed reliability among Arabic speakers [33,34] and was previously used in the CHART study, a cohort study that investigated the changes in health among Syrian refugees along their migration trajectories to Norway [5]. The question relating to the past in Syria was rephrased to describe a time point in the past by adding "when you were living in your home country". Participants could rate their health on scale from "very poor" (1), to "poor" (2), "neither" (3), "good" (4), and "very good" (5); the option "I do not know" (6) was also available. We then grouped "very poor/poor" and "very good/good".

Health literacy was measured with a self-assessment scale comprising three questions corresponding to the set of brief screening questions developed to detect inadequate or marginal health literacy in clinical settings [35]. The questions were: (1) "How often do you need someone helping you to read instruction pamphlets", (2) "How confident are you filling out a medical form by yourself", and (3) "How often do you have problems in learning about medical condition because of difficulty understanding written information?" Participants could answer the questions on a scale with the following five options: "never", "rarely/occasionally", "sometimes/somewhat", "often/quite a bit", and "always/extremely". We assigned zero (highest problems with reading or learning/not at all confident in filling out medical forms) to four points (no problems with reading or learning/extremely confident in filling out medical forms) to the scaled responses for the three questions. For instance, a participant was assigned 0 to the first question if the given response was "always" or 4 if the response was "never"; the score 2 would have been assigned if the response to the same question was "sometimes". The same scoring principle applied to questions 2 and 3 on the health literacy scale. The scores for each question were then summed to obtain a 0- or 12-point scale, which was categorized into low (score 0–5), medium (score 6–9), and high health literacy (score 10–12), as in prior research [36].

Sociodemographic and lifestyle factors were measured using questions adapted from prior studies in Norway [5,31,32]. These included gender, age, marital status, number of children, language, and ethnicity. Level of education was measured according to the Syrian educational system and grouped into basic (1–9 years), vocational (10–15 years), or

college/university (16 years and above). Past occupational status in Syria and current in Norway was grouped into "working", "not working" or "student".

Current lifestyle and health habits in Norway included smoking and alcohol use, frequency of physical activity, weight, and height. Based on the latter variables, we calculated body mass index (BMI) to be categorized into underweight, normal weight, overweight, or obese.

### *4.3. Data Analysis*

We quality checked the data for the presence of duplicate responses and unreliable answers. We considered unreliable responses to the electronic questionnaire where the time of completion was ≤5 min. Descriptive statistics were carried out overall and by sex. To compare differences in the proportions of prescription-only medication use among the same participant at two point of times in life, namely before and after the arrival to Norway, we used the McNemar test. Differences in proportions of medication use by sex and health literacy were examined via the chi-square or Fisher exact test. A *p*-value < 0.05 was considered statistically significant in this analysis. Data were processed and analyzed using IBM SPSS version 26.0 and Stata MP version 16.

### 5. Conclusions

In a population of immigrants from Syria living in western Norway, there was a relatively high use of medications in both Norway and Syria, particularly painkillers and analgesics, which deserves some attention. Even though past use of medication without medical prescription was very common in Syria, this declined substantially in Norway, reaching a prevalence estimate comparable to that observed in the hosting community. The use of prescription-only medications only in Norway was uncommon, while one every seven participants reported such use only in Syria. Low health literacy was related to the greater use of antibiotics and total medication use in the homeland Syria but not in the hosting country. Our study has a small number of participants, and this limitation must be considered when interpreting the findings. Nationwide research with a larger sample is needed to better understand the health-related challenges faced by immigrant groups in Norway and the factors associated with changes in medication use in the transition from Syria to Norway, including the effects of targeted health literacy interventions.

**Supplementary Materials:** The following supporting information can be downloaded at: https://www.mdpi.com/article/10.3390/pharma1020008/s1, Figure S1: Proportion of past medication use without medical prescriptions in Syria, by sex; Figure S2: Proportion of recent prescription-only medication use in Norway, by sex and frequency of use; Figure S3: Proportion of past medication use without medical prescriptions in Syria, by level of health literacy (low, medium, high); Figure S4: Proportion of recent prescription-only medication use in Norway, by level of health literacy (low, medium, high); Figure S5: Proportion of recent OTC medication use in Norway, by level of health literacy (low, medium, high); Table S1: Proportion of recent OTC medication use in Norway, by sex; Supplement S1: The study questionnaire.

**Author Contributions:** Conceptualization, E.D., S.H., G.D. and A.L.; methodology, A.L. and G.D.; formal analysis, G.D. and A.L., data curation, G.D.; writing—original draft preparation, G.D. and A.L.; writing—review and editing, E.D., S.H., G.D. and A.L.; supervision, A.L. All authors have read and agreed to the published version of the manuscript.

**Funding:** This research received no external funding.

**Institutional Review Board Statement:** The study was conducted in accordance with the Declaration of Helsinki. We applied for ethical approval to the Norwegian Regional Committees for Medical and Health Research Ethics, South-East (REK sør-øst) (Reference: 32352). REK concluded that ethical approval was not required for this study. The Norwegian Centre for Research Data (reference: 584985) approved the study.

**Informed Consent Statement:** Informed consent, digital or written, was obtained from all subjects involved in the study. Before giving informed consent and thereby accessing the questionnaire, all participants had to read the study description and aim. This work was performed on the TSD (Tjeneste for Sensitive Data) facilities, owned by the University of Oslo, operated and developed by the TSD service group at the University of Oslo, IT-Department (USIT).

**Data Availability Statement:** Data is contained within the article and supplementary material.

**Acknowledgments:** The authors would like to express our gratitude to participants in the study, the refugee center, and the pharmacy for their invaluable help in data collection. This work was performed on the TSD (Tjeneste for Sensitive Data) facilities, owned by the University of Oslo, operated and developed by the TSD service group at the University of Oslo, IT-Department (USIT). (tsd-drift@usit.uio.no). We thank the Norwegian Society for Pharmacoepidemiology for granting a master student scholarship to G.D.

**Conflicts of Interest:** The authors declare no conflict of interest.

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
