# Peer review of "Medication Use among Immigrants from Syria Living in Western Norway: A Cross-Sectional Study"

_2813-0618, doi:10.3390/pharma1020008_

Round 1

Reviewer 1 Report

The new information of manuscript  is very few comparing to the reference 5 - Strømme EM, Haj-Younes J, Hasha W, Fadnes LT, Kumar B, Igland J, Diaz E: Health status and use of 272 medication and their association with migration related exposures among Syrian refugees in Lebanon and 273 Norway: a cross-sectional study. BMC Public Health 2020, 20(1):34 -.

The European drug market is well regulated, so  the difference of AB consumption is natural.

The investigated number of  immigrants is low, the interpretation of data is inappropiate

Both method and interpreataton are weak.

The conclusion is correct, but is the same as was of the mentioned  earlier publication, without any additional suggestion.

Author Response

We thank the Reviewer for the valuable feedback provided. Please find below our point-by-point responses to the comments. We have split the comments in different points to facilitate readability.

Comment 1: The new information of manuscript  is very few comparing to the reference 5 - Strømme EM, Haj-Younes J, Hasha W, Fadnes LT, Kumar B, Igland J, Diaz E: Health status and use of 272 medication and their association with migration related exposures among Syrian refugees in Lebanon and 273 Norway: a cross-sectional study. BMC Public Health 2020, 20(1):34 -.

Response 1: Thank you for raising this comment. The study by Strømme et al. is indeed a comprehensive and well-conducted study about medication use and health status among Syrian refuges in a transit setting and in the host country. However, the above study did not explicitly indicate the positive or negative difference in proportion of different medications in the transition from Syria to Norway in the same individual, overall and according to sex. This is an aspect we have attempted to elaborate on in our work. In addition, the study by Stømmen did not examine use of antibiotics in this population, while we attempted to capture and measure this information.

To provide more novel aspects into the article, we have now added new results on health literacy of participants, and how health literacy is related to reporting medication use in Syria and Norway. This factor was not measured in the study by Strømmen et al., so we hope the Reviewer will find this addition valuable.

New results on the distribution of health literacy in the study population are described in Table 1. The extent of medication use (prescription-only and OTC) in Syria and Norway are now presented in Figures S4-S6.

Comment 2: The European drug market is well regulated, so  the difference of AB consumption is natural.

Response 2: We totally agree with the Reviewer that differences in drug market and regulations do exist in Syria versus Norway, and these can well explain the observed differences. However, multiple studies have raised concerns about inappropriate use of drugs, especially antibiotics, among immigrants because of the extensive self-medication behavior and possibility to purchase prescription-only medicines without a prescription at the pharmacies [1]. We believe that our quantification of the reduction in use of antibiotics in the transition from Syria to Norway among the same individual is important result which is to date lacking in Norway. This new knowledge is important given the increasing concerns about antibiotic resistance in Europe and worldwide, and addresses some of the concern of the Norwegian government that are now explained in the Introduction (see Response no. 1 to the Academic Editor for details).

Our new results also show that low health literacy is related to greater antibiotic use, both only in the homeland Syria, not in Norway. This further corroborates the role of stricter regulations and better structured healthcare system in the hosting country.

Comment 3: The investigated number of immigrants is low, the interpretation of data is inappropriate. Both method and interpreataton are weak. The conclusion is correct, but is the same as was of the mentioned  earlier publication, without any additional suggestion.

Response 3: We acknowledge that the number of participants is low in our study, and have further amended the Limitation section (as also requested by Reviewer 2 and the Academic Editor). At the same time – as also stated in the title – the study was conducted in a specific area of Norway and is not nationwide. The Limitation section now reads: “One important limitation of this study is the small sample size; the low number of participants impeded us to ex-plore more closely use of those medications that are less commonly used (e.g., drugs for diabetes, psychotropics), as well as to examine predictors of change in prescription-only medication from Syria to Norway”; “Furthermore, interpretation and generalizability of our results must be carefully considered in view of the small number of participants recruited.” 

We have interpreted our results in view of the limitation of small sample size, and we do not present data where the number of participants is too low (e.g., for medication groups with very low number of users) in view of this.

We respectfully disagree with the Reviewer that our method is weak, as respondents were recruited from different sites (the service center for refugees, and a centrally located community pharmacy) to minimize risk of selection bias. The questionnaire was available in Arabic language as paper-and-pencil and electronic format, to accommodate all needs and skills of participants. Further, the questionnaire included previously validated measures and questions. All these measures at the phase of study design were applied to increase study validity and minimize risk of multiple biases. If the Reviewer has specific concerns about the methodology, we are happy to address them if more specific details are given.

As described in Response no. 1, we have now added additional results on the role of health literacy, and amended the Conclusion with the following text “Low health literacy was related to greater use of antibiotics and total medication use in the homeland Syria, but not in the hosting country. Nation-wide research of greater sample size is needed to better understand health-related challenges faced by immigrant groups in Norway and factors associated with change in medication use in the transition from Syria to Norway, including the effect of targeted health literacy interventions”.

References

  1. Khalifeh MM, Moore ND, Salameh PR: Self-medication misuse in the Middle East: a systematic literature review. Pharmacol Res Perspect 2017, 5(4).

Reviewer 2 Report

I would like to thank the authors for their work.

This paper aims to quantify and compare the use and change in use of prescription-only medications purchased in the past in Syria without medical prescription versus nowadays in Norway after medical prescription, and to examine use of any medication as well as over-the-counter (OTC)  medications, overall and according to sex in the home and host country.

Before publication I would like to suggest some minor changes.

1) Abstract section, page 1, line 20: I would like to highlight a typo "78%.4" instead of "78.4%";

2) Material and Methods section: I suggest to move this section after the introduction;

3) Was the questionnaire checked by a native Arabic speaker?

4) Statistical analysis: have you assessed the distribution of continuous variables? Why the analysis have been performed with two different softwares (SPSS and Stata)?

4) Given the inevitable biases associated with the study design (and well reported, thank you for that) I would suggest toning down the conclusions a bit

Author Response

We thank the Reviewer for the valuable feedback provided. Please find below our responses your comments. We have split the comments in different points to facilitate readability.

Comment 1: I would like to thank the authors for their work.

This paper aims to quantify and compare the use and change in use of prescription-only medications purchased in the past in Syria without medical prescription versus nowadays in Norway after medical prescription, and to examine use of any medication as well as over-the-counter (OTC)  medications, overall and according to sex in the home and host country.

Response 1: Thank you for the positive feedback.

Comment 2: Abstract section, page 1, line 20: I would like to highlight a typo "78%.4" instead of "78.4%";

Response 2: Thank you for noticing this typo, we have now corrected it.

Comment 3: Material and Methods section: I suggest to move this section after the introduction;

Response 3: We have followed the template of the journal, which places the Material and Methods section at the end of the manuscript. We agree with the Reviewer that is not usual way of structuring the article, but that is required by the journal.

Comment 4: Was the questionnaire checked by a native Arabic speaker?

Response 4: Thank you for this comment. Yes, the questionnaire was translated by a native Arabic speaker from Syria (the first author). We have now specified this detail in the Methods, which reads: “The questionnaire was first developed in English, and then translated into Arabic by a native Arabic speaker from Syria”.  

Comment 5: Statistical analysis: have you assessed the distribution of continuous variables? Why the analysis have been performed with two different softwares (SPSS and Stata)?

Response 5: We did not include continuous variables in the analysis, they were all defined as categorical. We used both SPSS and Stata because the last author quality checked the data analysis (in Stata) done by the first author (in SPPS). Further, some results, e.g. the 95% CI of the difference in proportion in the McNemar test, could only be estimated by using Stata.

Comment 5: Given the inevitable biases associated with the study design (and well reported, thank you for that) I would suggest toning down the conclusions a bit

Response 5: thank you for this comment. We have toned down our Conclusions in light of the study limitation by adding this statement: “Our study has a small number of participants, and this limitation must be considered when interpreting the findings”.

Round 2

Reviewer 1 Report

The title of figures should not be separate! That kind of structure makes confusion of understandind!

On figure 3 the group name "spray -drops"  should be changed  - thay are pharmaceutical forms, but the others are pharmaco-therapeutic nomenclature, 

The method of health literacy measurement is not clear enough, please give more details regarding the scoring & grading!

Author Response

We thank the Reviewer for the additional feedback. Please find our number replies below. 

Note 1: The title of figures should not be separate! That kind of structure makes confusion of understandind!

Reply 1: We have embedded the supplementary figures in the word document with supplementary material along with the corresponding titles and footnotes. We additionally provide supplementary figures as separated figure files, so that the Editorial office has access to best formatted file format. 

Note 2: On figure 3 the group name "spray -drops"  should be changed  - thay are pharmaceutical forms, but the others are pharmaco-therapeutic nomenclature, 

Reply 2: We have now renamed "spray/drops" as "decongestant nasal spray/drops" in now Figure S5 and Table S1.  Please note that we originally presented recent use of OTC drugs in Norway as figure, but we have now presented same data in the format of a Table (Table S1). This revision was motivated by the fact that in males there was no reported daily use of two medications, and so the table can better illustrate this then the original figure. The manuscript has been edited by providing the correct reference to these supplementary figures and table. 

Note 3: The method of health literacy measurement is not clear enough, please give more details regarding the scoring & grading!

Reply 3: We have added additional details about the measure of health literacy in the Methods section.